# Scores for sepsis detection and risk stratification – construction of a novel score using a statistical approach and validation of RETTS

**Lisa Mellhammar**[1]\*, **Adam Linder**[1], **Jonas Tverring**[1], **Bertil Christensson**[1], **John H. Boyd**[2], **Per Åkesson**[1], **Fredrik Kahn**[1]

**1** Division of Infection Medicine, Department of Clinical Sciences, Lund University, Lund, Sweden, **2** Division of Critical Care Medicine , Centre for Heart Lung Innovation, St. Paul's Hospital, University of British Columbia, Vancouver, BC, Canada

\* lisa.mellhammar@med.lu.se

**Data Availability Statement:** The data in the study is based on patient material and since we still have a code key it is under the GDPR only considered to

## Abstract

### Background

To allow early identification of patients at risk of sepsis in the emergency department (ED), a variety of risk stratification scores and/or triage systems are used. The first aim of this study was to develop a risk stratification score for sepsis based upon vital signs and biomarkers using a statistical approach. Second, we aimed to validate the Rapid Emergency Triage and Treatment System (RETTS) for sepsis. RETTS combines vital signs with symptoms for risk stratification.

### Methods

We retrospectively analysed data from two prospective, observational, multicentre cohorts of patients from studies of biomarkers in ED. A candidate risk stratification score called Sepsis Heparin-binding protein-based Early Warning Score (SHEWS) was constructed using the Least Absolute Shrinkage and Selector Operator (LASSO) method. SHEWS and RETTS were compared to National Early Warning Score 2 (NEWS2) for infection-related organ dysfunction, intensive care or death within the first 72h after admission (i.e. sepsis).

### Results

506 patients with a diagnosed infection constituted cohort A, in which SHEWS was derived and RETTS was validated. 435 patients constituted cohort B of whom 184 had a diagnosed infection where both scores were validated. In both cohorts (A and B), AUC for infection-related organ dysfunction, intensive care or death was higher for NEWS2, 0.80 (95% CI 0.76–0.84) and 0.69 (95% CI 0.63–0.74), than RETTS, 0.74 (95% CI 0.70–0.79) and 0.55 (95% CI 0.49–0.60), $p = 0.05$ and $p < 0.01$, respectively. SHEWS had the highest AUC, 0.73 (95% CI 0.68–0.79) $p = 0.32$ in cohort B.

be pseudo-anonymized and not de-identified. Furthermore, since there are many individual variables and sensitive patient information, there is a possibility that patients might be identified due to their comorbidities, patient characteristics and time of encounter. Hence, we were not granted an ethical permit to publish individual data but merely publishing data on group level. This is also clearly stated in the information to patients which the participants have signed. We considered uploading the full data set for publication but due to the ethical restrictions imposed on us, this is unfortunately not possible to do. In order to assure compliance with the information given to the patients, data can be shared upon request from ethics committee registrator@etikprovning.se, 0046104750800 (Swedish Ethical Review Authority) and from Cantonal Ethics Committee Zurich info.kek@kek.zh.ch.

**Funding:** Swedish Government Funds for Clinical Research (ALF), the Crafoord foundation, the Swedish Society of Medicine, the Thelma Zoégas foundation, the foundation of Apotekare Hedberg, the foundation of Magnus Bergvall, the Royal Physiographic Society, Lund, the Foundations of Skåne University Hospital, the foundation of Alfred Österlund and, the foundation of Clas Groschinsky The funders had no role in study design, data collection and analysis, decision to publish, or preparation of the manuscript.

**Competing interests:** Bertil Christensson, Per. Åkesson, and Adam Linder are listed as inventors on a patent on the use of HBP as a diagnostic tool in sepsis filed by Hansa Medical AB WO2008151808A1. This does not alter our adherence to PLOS ONE policies on sharing data and materials. All other authors have declared no relevant conflicts of interest.

**Abbreviations:** AUC, Area Under receiver operating characteristic Curve; CI, Confidence Interval; DBP, Diastolic Blood Pressure; ED, Emergency Department; ESS, Emergency Signs and Symptoms; ELISA, Enzyme-Linked ImmunoSorbent Assay; FiO$_2$, Fraction of Inspired Oxygen; HBP, Heparin-Binding Protein; LASSO, Least Absolute Shrinkage and Selector Operator; LOWESS, Locally WEighted Scatterplot Smoothing; NEWS2, National Early Warning Score 2; OR, Odds Ratio; PaO$_2$, Partial pressure of Oxygen; qSOFA, Quick Sequential Organ Failure Assessment; RETTS, Rapid Emergency Triage and Treatment System; SaO$_2$, Oxygen Saturation; SE, Standard Error; SEWS, Sepsis Early Warning Score; SHEWS, Sepsis Heparin binding protein-based Early Warning Score; SBP, Systolic Blood

## Conclusions

Even with a statistical approach, we could not construct better risk stratification scores for sepsis than NEWS2. RETTS was inferior to NEWS2 for screening for sepsis.

## Introduction

Sepsis is a medical emergency, requiring early recognition and care. Its clinical features can vary and be vague and therefore difficult to detect. Sepsis is common, especially at Emergency Departments (ED), where it constitutes between 2–13% of encounters [1, 2].

Various risk stratification tools have been used in order to recognize and prioritize patients with risk of progression to sepsis. Any sepsis scoring system must be very sensitive, as this disease has both a high mortality rate and delayed treatment dramatically worsens outcome [3]. The National Early Warning Score 2 (NEWS2) is a modification of National Early Warning Score (NEWS) and is a risk stratification score for the probability of clinical deterioration of for example development of sepsis [4]. Although NEWS2 has the best accuracy for sepsis detection of the commonly used risk stratification scores, we have previously shown that a substantial portion of patients with sepsis goes undetected with a cut-off of NEWS2 $\geq$5 [5].

Risk stratification scores for deterioration have most commonly been constructed from analysis of the most abnormal vital signs in a given observation period [4, 6, 7]. Rapid Emergency Triage and Treatment System (RETTS) builds upon a vital sign based score and is widely used for triage at EDs in Sweden [7, 8]. RETTS uses a four-graded scale describing the levels of abnormal vital signs in combination with scores assigned for common Emergency Signs and Symptoms (ESS). RETTS is used to assign acceptable wait times before physician assessment, with 'red' the highest priority (S1 Table) [7].

More sophisticated but with greater complexity is to include biomarkers in the risk stratification scores. Lactate has been tested for inclusion in quick sequential organ failure assessment (qSOFA), but appears to offer no benefit [6]. Heparin-Binding Protein (HBP), another potential biomarker, is a granule protein which is released by neutrophils in response to bacterial products and neutrophil adhesion [9–13]. It has been found to be a superior biomarker than lactate to predict the development of sepsis in the ED [14].

We wanted to explore different approaches for sepsis risk stratification tools i.e. a statistical approach and the inclusion of biomarkers and symptoms in risk stratification tools.

By using two prospective, observational, multicentre cohorts of patients with blood drawn for biomarkers at presentation in an ED, the aims of this study were to a) develop a sepsis risk stratification tool based on the most predictive, minimal set of vital signs, lactate and HBP plasma levels and b) validate RETTS as able to predict both sepsis and subsequent 30-day mortality.

## Materials and methods

### Ethics

Ethical approvals were obtained from the regional ethical board in Lund (approval number 2010/205 and 2014/4), the regional ethical board in Bern (approval number KEK 315/14) and the regional ethical board in Vancouver (approval number H11-00505). All included patients gave written informed consent in cohort A. The study is in accordance with the approval and the informed consents. The study is also in accordance with the approvals and informed

Pressure; SIRS, Systemic Inflammatory Response Syndrome.

consents for cohort B. In cohort B, included patients all gave written informed consent or, if unable to give informed consent, next-of-kin was asked for permission. For patients that died without being able to leave informed consent, the use of data and samples was requested at the local ethics committee.

## Patients

**Cohort A (Suspected infection in the ED).** Data from an observational, multicentre convenience trial of biomarkers were used. Patients were included prospectively between 2011–2012 at EDs. The study has been described in detail elsewhere [14]. Patients at the Swedish sites were included in this analysis.

In summary, patients ≥18 years with a suspected infection and at least one of Systemic Inflammatory Response Syndrome (SIRS) criteria or self-reported fever or chills, were included at presentation.

The following data were registered at enrolment: data on demography, comorbid conditions, medication and vital signs. Samples for laboratory testing were ordered. Retrospectively, data on organ dysfunction, treatment, intensive care, infection diagnosis and 30-day mortality were gathered from medical records and a national death registry.

**Cohort B (acutely ill undifferentiated ED patients).** Data from a multicentre, observational, convenience trial of sepsis biomarkers were used. Between 2015–2016, patients were enrolled at EDs in the study that has been described in detail elsewhere [15]. Patients who fulfilled at least one of the following requirements, were included: Respiratory rate >25 breaths per minute, heart rate >120 beats per minute, altered mental awareness, systolic blood pressure (SBP) below 100 mmHg, oxygen saturation ($SaO_2$) <90%, or <93% if ongoing treatment with oxygen. Both infected and non-infected patients were included. Data on demography, comorbid conditions, medication and vital signs were registered at enrolment and samples for laboratory testing were ordered. Retrospectively, data on organ dysfunction, treatment, intensive care, infection diagnosis and 30-day mortality were gathered from medical records and a national death registry.

At inclusion, patients from the Swedish sites in cohort B were categorized according to the ESS algorithms for RETTS. Patients from the Swedish sites were also followed up for 30-day mortality.

## Definitions

Sepsis was defined as a probable or verified infection based on clinical presentation, laboratory results, microbiological samples and radiologic examinations, and an acute organ dysfunction of no other apparent or pre-existing cause. We applied the organ dysfunction definitions from the former sepsis-2 definition since this was consensus criteria at the time the data were collected, although we did not require 2 SIRS criteria due to its lack of validity [16, 17]. For the definitions of organ dysfunction see S2 Table. Patients with infection that died or were treated at the intensive care unit within 72 hours were also regarded as suffering from sepsis. We applied this combined outcome since it is most probably sepsis that causes death or intensive care in infected patients but it can potentially be undetected. This combined outcome will hereafter be referred to as sepsis. To validate the combined outcome for sepsis, we also analysed risk stratification scores for predicting a maximal rise in SOFA score of 2 or more within 72h from admission together with infection (Sepsis-3 definition) in cohort B. However, in cohort A SOFA score was only available at baseline. Hence, the validation of the combined outcome for sepsis was made for the presence of sepsis at inclusion in this cohort.

For PaO$_2$/FiO$_2$ ratio for patients with SaO$_2$ 90–94% and for patients with COPD and SaO$_2$ 87–95% and simultaneous oxygen supply in cohort B, the Severinghaus equation was used [18].

Since acute neurological dysfunction was direct part of scores being evaluated, it was excluded as an organ dysfunction defining sepsis.

RETTS was defined according to S1 Table. ESS47 covers infection and categorize patients as red if presenting with petechiae and concomitant signs of infection.

NEWS2 was used as reference risk stratification score since it has the best accuracy for sepsis detection of the risk stratification scores widely used and validated (S3 Table)[5].

Biomarkers included in the construction of a candidate risk stratification tool were lactate and HBP. The biomarkers lactate and HBP were selected for their prognostic abilities in sepsis and the availability of point-of-care testing [14, 15]. Although in the study, HBP was analysed with Enzyme-Linked ImmunoSorbent Assay (ELISA) at a centralized laboratory. Lactate and blood sample analyses used for organ dysfunction were analysed at the clinical chemistry departments at each hospital.

## Statistical methods

The least absolute shrinkage and selector operator (LASSO) method was used for construction of a candidate risk stratification tool. The LASSO method avoids correlating covariates from being included in a prediction model [19, 20]. LASSO was preceded by locally weighted scatterplot smoothing (LOWESS) regression for eligible parameters with reference to the outcome, sepsis including admittance to intensive care due to an infection or infection-related mortality within 72 hours from enrolment. This generated a smooth curve for selection of intervals for parameters to include in the LASSO regressions. Variables were dichotomized within the selected intervals and entered to the LASSO regressions. The LASSO included a 10-fold cross-validation with areas under the receiver operating characteristic curves (AUC) optimization and was iterated 50 times. All values optimizing AUC in more than 50% of the LASSO analyses and with a coefficient $\geq$0.05 were then entered in to a second set of LASSO regressions, unless the values were adjacent. If adjacent values, the one with the higher coefficient was chosen. Values included in more than 50% of the second set of LASSO regressions and with a coefficient of $\geq$0.05 within 1 standard error (SE) from max AUC were selected. These values were given a score proportional to their coefficients generated by the second set of LASSO regression and rounded to the closest integer. The cut-offs for these scores were set to also require scores from more than one parameter.

AUC, sensitivity, specificity and their 95% confidence interval (CI) were calculated. Odds ratio (OR) was calculated for the risk stratification scores relation to 30-day mortality. P-values were calculated with Chi$^2$-test when comparing proportions and using the formula of Delong for comparison of AUC. P-values below 0.05 were regarded as significant.

Patients with missing values, among the vital signs or biomarkers included in the risk stratification scores or included in construction of a risk stratification score, were excluded in the primary analyses.

Multiple imputation of missing values that are part of the risk stratification scores were executed using predictive mean matching and logistic regression with 20 imputation sets and the performances of different risk stratification scores were calculated in a sensitivity analysis.

The performances of risk stratification scores were analysed in imputed data sets. AUC's were calculated as medians of the pooled data and 95% CI's for all imputed datasets.

Analyses were performed using glmnet package, R version 3.4.0 (The R Foundation for Statistical Computing) and SPSS software system version 23.0 (IBM, Armonk, NY).

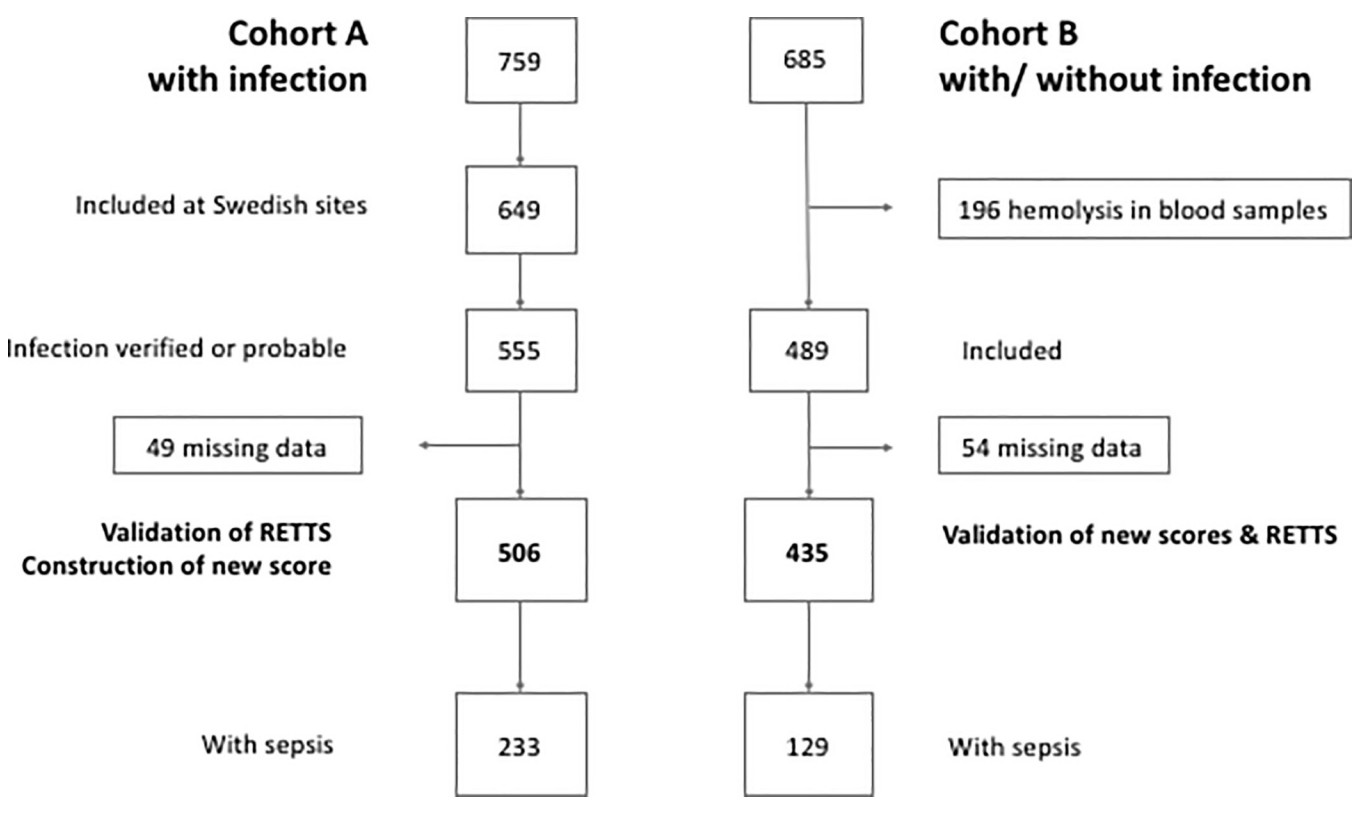

**Fig 1. Flow chart.** RETTS (Rapid Emergency Triage and Treatment System).

## Results

### Cohort A

Patient inclusion, exclusion and classification as sepsis is outlined in Fig 1. A total of 506 patients with an infection and complete data on vital signs and laboratory analyses were included in the primary analyses in cohort A. 283 (56%) had at least one comorbidity, 12 (5%) were admitted to ICU and 4 (2%) died. 233 fulfilled the combined outcome referred to as sepsis (infection-related organ dysfunction, intensive care or death within the first 72h after admission). Of the 233 patients with the combined outcome, 228 had infection-related organ dysfunction, 4 were treated in the ICU and one died without organ dysfunction being registered. The two independent infectious disease physicians who reviewed the data attributed the death and the ICU-care for these five patients to infectious diseases.

Patient characteristics are presented in Table 1. Table 2 compares RETTS to NEWS2 for sepsis. RETTS had lower AUC than NEWS2 for sepsis detection, 0.74 (95% CI 0.70–0.79) and 0.80 (95% CI 0.76–0.84), respectively, $p = 0.05$.

### Construction of a new risk stratification score

A LOWESS regression for eligible parameters with reference to the composite outcome (S1 Fig) resulted in the following values included in the first set of LASSO regressions as ordinal variables: 5-year-age groups between 40 and 90, heart rate from 60 to 140 in groups of five following frequencies, SBP from 70 to 120 in groups of two following mmHg, diastolic blood pressure (DBP) from 40 to 90 in groups of two following mmHg, respiratory frequencies from

**Table 1. Patient characteristics.**

| | Cohort A | | | Cohort B | | |
|---|---|---|---|---|---|---|
| | Without sepsis n = 273 | With sepsis n = 233 | *p* | Without sepsis n = 306 | With sepsis n = 129 | *p* |
| Age, median | 52 | 70 | | 72 | 77 | |
| Female, n (%) | 106 (39) | 118 (51) | | 142 (46) | 64 (50) | |
| Comorbidities n (%) | | | | | | |
| Diabetes | 29 (11) | 45 (19) | <0.01 | 54 (18) | 33 (26) | 0.06 |
| Cardiovascular disease | 31 (11) | 78 (33) | <0.01 | 146 (48) | 71 (55) | 0.16 |
| Renal Disease | 15 (5) | 30 (13) | <0.01 | 32 (10) | 22 (17) | 0.06 |
| Liver Disease | 3 (1) | 1 (0) | 0.40 | 6 (2) | 7 (5) | 0.05 |
| Malignancy | 20 (7) | 24 (10) | 0.24 | 46 (15) | 17 (13) | 0.62 |
| Immunodeficiency | 9 (3) | 9 (4) | 0.73 | 3 (1) | 3 (2) | 0.27 |
| Respiratory Disease | 21 (8) | 32 (14) | 0.03 | 69 (23) | 38 (29) | 0.13 |
| No comorbidities | 126 (46) | 97 (42) | 0.31 | 103 (34) | 23 (18) | |
| Organ dysfunction, n (%) | | | | * | | |
| No organ dysfunction | | 5 (2) | | 154 (50) | 0 (0) | |
| Neurologic | | 37 (16) | | 64 (21) | 32 (25) | |
| Cardiovascular | | 186 (80) | | 80 (26) | 83 (64) | |
| Respiratory | | 61 (26) | | 59 (19) | 81 (63) | |
| Renal | | 25 (11) | | 52 (17) | 25 (19) | |
| Hematological | | 22 (9) | | 12 (4) | 12 (9) | |
| Hepatic | | 7 (3) | | 5 (2) | 4 (3) | |
| Intensive Care n (%) | | 12 (5) | | 20 (7) | 12 (9) | |
| 3-days mortality n (%) | | 4 (2) | | 4 (1) | 13 (10) | |

* Organ dysfunction without infection

20 to 40 in groups of two following rates and mental status classified as ordinal groups 1–5 according to whether normal, agitated, confused, drowsy or unconscious.

The LASSO regression was cross-validated and repeated with values dominating the first set of LASSO. The second set of LASSO regressions generated values that were given a score proportional to their coefficients. This resulted in construction of a new risk stratification scores (Tables 3 and 4) called Sepsis Early Warning Score (SEWS) for the model without HBP, and Sepsis HBP-based Early Warning Score (SHEWS) for a model with HBP. The cut-off values for risk of sepsis were set at 7 points for SEWS and 10 points for SHEWS.

## Sensitivity analyses

For 49 patients excluded with missing data, multiple imputation was performed and validation for RETTS was repeated in the imputed data sets. For missing data and demographics and outcome of patients with missing data, see S4 and S5 Tables.

Variables imputed were systolic blood pressure, diastolic blood pressure, heart frequency, respiratory frequency, temperature, mental status, $SaO_2$, oxygen treatment, lactate, age and HBP. Other parameters in the imputation although not imputed were comorbidities and outcome. Predictive Mean Matching were used for multiple imputation and logistic regression for binary variables. The models were validated by plots of imputations and iterations.

Data were assumed to be missing at random, conditional on observed data in the imputation model.

For results of analyses using multiple imputation, see S6 Table. AUC's were calculated as medians of the pooled data and 95% CI's for all imputed datasets.

**Table 2. Accuracy of risk stratification scores for sepsis, 95% CI within brackets.**

| | RETTS RED | RETTS <RED | NEWS2 ≥5 | NEWS2 <5 | SEWS ≥7 | SEWS <7 | SHEWS ≥10 | SHEWS <10 |
|---|---|---|---|---|---|---|---|---|
| **Cohort A** | | | | | | | | |
| **With sepsis** | 66 | 167 | 152 | 81 | * | * | * | * |
| **Without sepsis** | 12 | 261 | 55 | 219 | * | * | * | * |
| **Sensitivity** | 28 (23–35) | | 65 (59–71) | | * | * | * | * |
| **Specificity** | 95 (92–98) | | 80 (75–85) | | * | * | * | * |
| **AUC** | 0.74 (0.70–0.79) | | 0.80 (0.76–0.84) | | * | * | * | * |
| *p* (NEWS2) | 0.05 | | reference | | | | | |
| **30-day mortality** | 4 | 10 | 10 | 4 | 12 | 2 | 11 | 3 |
| **30-day survival** | 74 | 418 | 197 | 295 | 146 | 346 | 257 | 235 |
| **OR 30-day mort** | 2.3 (0.7–7.4) | | 3.7 (1.2–12.1) | | 14.2 (3.1–64.3) | | 3.4 (0.9–12.2) | |
| **Cohort B** | | | | | | | | |
| **With sepsis** | 74 | 55 | 108 | 21 | 108 | 21 | 107 | 22 |
| **Without sepsis** | 149 | 157 | 198 | 108 | 212 | 94 | 162 | 144 |
| **Sensitivity** | 57 (48–66) | | 84 (76–90) | | 84 (76–90) | | 83 (75–89) | |
| **Specificity** | 51 (46–57) | | 35 (30–41) | | 31 (26–36) | | 47 (41–53) | |
| **AUC** | 0.55 (0.49–0.60) | | 0.69 (0.63–0.74) | | 0.67 (0.61–0.73) | | 0.73 (0.68–0.79) | |
| *p* (NEWS2) | <0.01 | | reference | | 0.63 | | 0.32 | |
| **30-day mortality** | 20 | 8 | 26 | 2 | 26 | 2 | 27 | 1 |
| **30-day survival** | 158 | 166 | 221 | 103 | 240 | 84 | 186 | 138 |
| **OR 30-day mort** | 2.6 (1.1–6.1) | | 6.1 (1.4–26.0) | | 4.5 (1.1–19.6) | | 20.0 (2.7–149.2) | |

* Not able to validate, derived in this cohort

the analyses using multiple imputation yielded similar estimates as the analyses of the original data.

The validation of the combined outcome for sepsis was made for the presence of sepsis in this cohort, by analyzing risk stratification scores for detecting the combined outcome at inclusion compared to a presumed rise in SOFA score of 2 or more and infection (Sepsis-3 definition) at inclusion. When comparing risk stratification scores for detecting the combined outcome at inclusion to sepsis-3 at inclusion there were no differences except for SHEWS which did not perform as well in detecting sepsis-3 (AUC 0.79) as in detecting the combined outcome (AUC 0.86) and was not significantly higher than AUC for RETTS (S7 Table).

When excluding the 5 patients included in the composite outcome for sepsis but without registered organ dysfunction, as not possible to classify, the results were not significantly changed , RETTS AUC 0.75 (95% 0.71–0.79), NEWS2 AUC 0.80 (95% CI 0.77–0.84).

**Table 3. SEWS, Early Warning Score.**

| | 1 | 2 | 3 | 4 | 5 |
|---|---|---|---|---|---|
| **Age** | >45 | | | >60 | >80 |
| **Mental Status** | | | Confused or drowsy | | |
| **Respiratory Frequency** | | | | | >24 |
| **SBP (mmHg)** | | | <106 | <100 | |
| **DBP (mmHg)** | <78 | <58 | | | |
| **Heart Rate** | | >110 | | | |

**Table 4. SHEWS, Sepsis Heparin binding protein-based Early Warning Score.**

|  | 1 | 2 | 3 | 4 | 5 | 6 | 7 | 8 |
|---|---|---|---|---|---|---|---|---|
| **Age** | >45 |  |  | >60 | >80 |  |  |  |
| **Mental Status** |  |  | Confused or drowsy |  |  |  |  |  |
| **Respiratory Frequency** |  |  |  |  | >24 |  |  |  |
| **SBP (mmHg)** |  | <106 |  |  |  | <100 |  |  |
| **DBP (mmHg)** | <78 |  | <56 |  |  |  |  |  |
| **Heart Rate** |  | >110 |  |  |  |  |  |  |
| **HBP (ng/mL)** |  |  | >26 |  |  | >30 | >48 | >54 |

## Cohort B

Data on patient characteristics are presented in Table 1. Of 435 patients, 184 (42%) had a diagnosed infection. 129 (30%) experienced sepsis within 72 hours from enrolment (Fig 1). All 129 patients categorized as sepsis had organ dysfunction and not only death within 72h or ICU without organ dysfunction being registered (composite outcome).

When compared to cohort A, these patients had more comorbidities 309 (71%) and were more often admitted to ICU 32 (7%) or died 17 (4%).

SEWS and SHEWS were evaluated for their ability to predict sepsis. For cross tabulations, sensitivity, specificity, AUC and OR see Table 2. The new score with HBP, SHEWS, yielded the highest AUC (0.73). The new score without HBP (SEWS) (AUC 0.67) was inferior to NEWS2 (AUC 0.69), although not significantly. RETTS (0.55) had the lowest AUC of the validated scores ($p < 0.01$).

The patients from the Swedish sites (n = 354) were also classified according to the ESS. The ESS algorithms are used in combination with vital signs in RETTS and can give patients a higher priority due to symptoms. Only one patient was newly classified as red RETTS due to the ESS algorithm for infection. Eleven patients were classified as red due to other causes than infection, most often dyspnéa or chest pain with new left bundle branch block, ST-elevation or widespread, sudden pain with vegetative symptoms or unconsciousness. Neither did affect RETTS' performance.

When validated among the sub group of (retrospectively diagnosed) infected patients (n = 182), the discriminating capacity of NEWS2 and SHEWS did not change significantly, AUC 0.72 (95% CI 0.64–0.80) and 0.74 (95% CI 0.66–0.82), but the AUC for RETTS rose to 0.61 (95% CI 0.52–0.70), which was still inferior to NEWS2, $p = 0.02$.

## Sensitivity analyses

As in cohort A, multiple imputation of missing data for 54 patients excluded with missing data were performed. The analysis of performance of the different scores for the primary outcome was repeated in the imputed data sets.

For missing data and demographics and outcome of patients with missing data, see S4 and S5 Tables.

Variables imputed were systolic blood pressure, diastolic blood pressure, heart frequency, respiratory frequency, temperature, mental status, SaO$_2$, oxygen treatment, lactate. Other parameters in the imputation although not imputed were comorbidities, outcome, age and HBP. Predictive Mean Matching were used for multiple imputation and logistic regression for binary variables. Data were assumed to be missing at random, conditional on observed data in the imputation model. The models were validated by plots of imputations and iterations.

AUC's were calculated as medians of the pooled data and 95% CI's for all imputed datasets.

The analyses using multiple imputation rendered similar result as the complete cases analyses, thus the imputation analysis did not change the relation of AUCs for RETTS, SHEWS, SEWS and NEWS2. For complete results see S6 Table.

To validate the combined outcome for sepsis, we also analysed risk stratification scores for predicting a presumed rise in SOFA score of 2 or more <72h together with infection (Sepsis-3 definition) in this cohort. When comparing risk stratification scores for predicting the combined outcome to sepsis-3 there were no differences except for NEWS2 which performed better in predicting sepsis-3 (AUC 0.79) compared to the combined outcome (AUC 0.69) (S8 Table).

## Discussion

We used a statistical approach in constructing a new risk stratification score for sepsis. The new score, SHEWS, had the highest accuracy for detection and prediction of sepsis in the ED, although not statistically superior to NEWS2. Both SHEWS and NEWS2 performed significantly better than RETTS for sepsis detection, even though RETTS combines vital signs with symptoms for risk stratification.

RETTS has had little previous validation for sepsis detection, but our results are in concordance with the previous studies. Askim *et al.* demonstrated a sensitivity of 34% and specificity of 95% for red RETTS for detecting severe sepsis, in a cohort of infected patients at ED [21]. In the present study, RETTS had a sensitivity of 28% and a specificity 95% in cohort A and a sensitivity of 57% and specificity of 51% among infected patients in cohort B. A study of RETTS' association with the final hospital diagnosis in children demonstrated sepsis to be the most frequent inappropriately classified, time-dependant condition [22].

Interestingly, when constructing a new risk stratification score for sepsis, the LASSO regression found, among others, exactly the same parameters and cut-off values that are included in the qSOFA score, with the exception of respiratory rate which differed slightly, $\geq 22$ and $> 24$ respectively [23]. This confirms the importance of the parameters and cut-off values selected for qSOFA in predicting sepsis, yet studies have indicated qSOFA to be too simplified [5, 6]. These parameters and cut-off values are also components of NEWS2 [4].

Perhaps it is not possible to reach higher AUC for sepsis recognition than NEWS2, using easy available vital signs and the biomarkers included in the statistical model. Other scores have not been able to demonstrate superiority over NEWS2 for prognostic accuracy for deterioration in infected patients [5, 24].

HBP increased the performances for the new risk stratification score, but we only used HBP and lactate as eligible biomarkers when constructing the new score. Other biomarkers for sepsis might have better additive effects.

One promising approach for a risk stratification score is the use of machine learning for continuous sampling of data and calculation of real-time scores. Targeted real-time early warning score has been demonstrated to perform well for both sepsis and septic shock and near real-time automated SOFA has proven to have a strong agreement with manual SOFA score calculation [25, 26]. Unfortunately, continuous sampling is hampered in the ED, due to the short observation period [25, 27].

Strengths of this study are the validation of the scores both among infected patients and among unselected patients at the ED with sepsis according to clinical assessment. A sepsis risk stratification score performs most likely better among infected patients, otherwise it is supposed to identify infection as well. However the initial assessment of whether the patient is infected or not has often proved to be wrong, why it is important to validate these scores among both infected and unselected ED patients [28].

A major limitation is that patients in the study that are considered as falsely classified as positive by the risk stratification scores, still can suffer from other time-critical conditions especially in the cohort which includes patients with infection as well as without. Also, the inclusion criteria for the cohort B are largely coherent with RETTS. These weaknesses might lead to RETTS being estimated as more sensitive but less specific. When analysing the sub cohort of infected patients in cohort B, the performance of RETTS was however not significantly changed. The low 30-day mortality resulted in difficulty to reliably assess the secondary outcome, to validate the risk stratification scores for 30-day mortality.

There were missing data at admission on variables for the validated risk stratification scores, although not a high proportion <7%. We performed multiple imputation as a sensitivity analysis to address this problem.

We assumed data to be missing at random. This is not testable, but becomes more reasonable within a model like ours that includes several characteristics, including predictors and the outcome. The probability that vital signs and laboratory values are missing is related to other parameters measured and hence missing at random is a valid assumption. Therefore, we can use multiple imputation to estimate the effects on the missing vital signs and laboratory values. Multiple imputation is commonly used when evaluating clinical risk scores [6, 29].

Another limitation is the use of the sepsis-2 definition. The proposed definition of sepsis-3 provoked a fierce discussion and the new definition has not by far been officially accepted by all professional associations. Even though we now believe sepsis-3 to be helpful for clinicians, it was not published at the time the data were collected and accordingly we use the sepsis-2 definition. We did however perform a sensitivity analysis which compared our combined outcome for sepsis to sepsis-3 in cohort B and for detection of sepsis at inclusion in cohort A. The sensitivity analysis did not change the results.

RETTS is commonly used in Sweden, but its external validity outside of Sweden is limited. However, we wanted to explore different approaches for sepsis risk stratification tools i.e. the addition of biomarkers and symptoms.

It is not evident which statistics best reflect the performance of risk stratification scores. Risk stratification or triage scores used for sepsis detection needs a high sensitivity since the consequences of delayed diagnosis are severe. The scores are simple and cheap, although the "cost" of a false positive score is the risk of another patient being lower prioritized for clinical evaluation. In this study the positive predictive value was at lowest one third, so the number needed to evaluate never exceeded 3. Hence, the sensitivity remains the crucial metrics in this study which was far too low for RETTS for sepsis detection. The other risk stratification score performed better, but there is still scope for improvement.

## Conclusion

Even with a statistical approach, we could not construct better risk stratification scores for sepsis than NEWS2. RETTS was inferior to NEWS2.

## Supporting information

**S1 Fig. LOWESS curves.**
(TIFF)

**S1 Table. RETTS.**
(DOCX)

**S2 Table. Organ dysfunction definition.**
(DOCX)

**S3 Table. NEWS2.**
(DOCX)

**S4 Table. Missing data.**
(DOCX)

**S5 Table. Demographics and outcome of patients with missing data.**
(DOCX)

**S6 Table. AUC for different risk stratification scores in multiple imputation datasets.**
(DOCX)

**S7 Table. AUC for different risk stratification scores detection of combined outcome for sepsis compared to sepsis-3 definition, cohort A.**
(DOCX)

**S8 Table. AUC for different risk stratification scores prediction of combined outcome for sepsis compared to sepsis-3 definition, cohort B.**
(DOCX)

## Acknowledgments

Parham Sendi for study support at Bern.

## Author Contributions

**Conceptualization:** Lisa Mellhammar, Adam Linder, Bertil Christensson, Per Åkesson, Fredrik Kahn.

**Data curation:** Lisa Mellhammar.

**Investigation:** Lisa Mellhammar, Adam Linder, Jonas Tverring, John H. Boyd, Per Åkesson, Fredrik Kahn.

**Methodology:** Lisa Mellhammar, Adam Linder, Bertil Christensson, Fredrik Kahn.

**Supervision:** Adam Linder, Bertil Christensson, Fredrik Kahn.

**Writing – original draft:** Lisa Mellhammar.

**Writing – review & editing:** Lisa Mellhammar, Adam Linder, Bertil Christensson, John H. Boyd, Per Åkesson, Fredrik Kahn.

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
