## [Decision Letter · Decision Letter 0]

18 Dec 2019

PONE-D-19-32440

Scores for sepsis detection and risk stratification – construction of a novel score using a statistical approach and validation of RETTS

PLOS ONE

Dear Dr Mellhammar,

Thank you for submitting your manuscript to PLOS ONE. After careful consideration, we feel that it has merit but does not fully meet PLOS ONE’s publication criteria as it currently stands. Therefore, we invite you to submit a revised version of the manuscript that addresses the points raised during the review process.

Thank you for this interesting and well-done submission. Overall, this paper addresses an important topic. I like the premise and the results, in the sense that they highlight that a more complex (in this case, biomarker-based) approach to prognostication is not necessarily superior simply because of the complexity.

In addition to comments raised by the reviewers, I have two questions/concerns related to assumptions made in methods section. The first (and more significant), is the part of the definition for sepsis, as stated in lines 157-159. I wonder if this definition is overly broad, in the sense that there are many reasons for death in 72 hours or treatment in the ICU, in the setting of "infection" that are not directly sepsis. I think that further justification for this definition is required (can the charts for patients who qualified as sepsis by these criteria be reviewed?); if this is not possible, the potential ramifications of this limitation need to be discussed.

My second concern, although related to a secondary outcome, is about the decision to use multiple imputation for missing data in the prediction scores--I wonder if the data is actually missing not at random (MNAR). Patients who are less ill may have fewer tests/labs ordered or had VS documented less frequently (or they may be repeated less often) and thus existing values may be more deranged as they represent the sicker spectrum of the population. As such, what data does exist may not be truly representative of the entire population thereby potentially biasing the results of the multiple imputation data sets; this would be particularly true for lactate in this case. Perhaps this could explain why the two analyses (with/without) missing data produced similar results?

We look forward to receiving your revised manuscript.

Kind regards,

Robert Ehrman, MD, MS

Academic Editor

PLOS ONE

Journal Requirements:

'the regional ethical board in Lund (approval number 2010/205 and 2014/4).

the regional ethical board in Bern (approval number KEK 315/14)

the regional ethical board in Vancouver (approval number H11-00505).'

a. Please amend your current ethics statement to confirm that your named institutional review board or ethics committee specifically approved this study.

3. In the ethics statement in the manuscript and in the online submission form, please provide additional information about the patient records used in your retrospective study.

Specifically, please ensure that you have discussed whether all data were fully anonymized before you accessed them and/or whether the IRB or ethics committee waived the requirement for informed consent.

If patients provided informed written consent to have data from their medical records used in research, please include this information.

4. Thank you for stating the following in the Competing Interests section: "Bertil Christensson, Per.Åkesson, and Adam Linder are listed as inventors on a patent on the use of HBP as a diagnostic tool in sepsis filed by Hansa Medical AB. All other authors have declared no relevant conflicts of interest."

We note that you have a patent relating to material pertinent to this article.

a. Please provide an amended statement of Competing Interests to declare this patent (with details including name and number), along with any other relevant declarations relating to employment, consultancy, patents, products in development or modified products etc. Please confirm that this does not alter your adherence to all PLOS ONE policies on sharing data and materials, as detailed online in our guide for authors http://journals.plos.org/plosone/s/competing-interests by including the following statement: "This does not alter our adherence to  PLOS ONE policies on sharing data and materials.” If there are restrictions on sharing of data and/or materials, please state these. Please note that we cannot proceed with consideration of your article until this information has been declared.

6. Please upload a new copy of your Supporting Information Figure as the detail is not clear. Please follow the link for more information: http://blogs.PLOS.org/everyone/2011/05/10/how-to-check-your-manuscript-image-quality-in-editorial-manager/

Reviewers' comments:

Reviewer's Responses to Questions

**Comments to the Author**

1. Is the manuscript technically sound, and do the data support the conclusions?

Reviewer #1: Yes

Reviewer #2: Partly

2. Has the statistical analysis been performed appropriately and rigorously? 

Reviewer #1: Yes

Reviewer #2: Yes

3. Have the authors made all data underlying the findings in their manuscript fully available?

Reviewer #1: Yes

Reviewer #2: No

4. Is the manuscript presented in an intelligible fashion and written in standard English?

Reviewer #1: Yes

Reviewer #2: Yes

5. Review Comments to the Author

Reviewer #1: The study utilized sophisticated statistical techniques to develop risk stratification model for sepsis. there are several concerns from me.

1. The score developed in the study requires lactate and HBP, which is difficult to obtain at the very beginning. in particular the HBP is not routinely measured. thus, the authors must discuss that the applicability of the model is limited. lactate is not routinely measured for infection but suspected sepsis.

2. "Patients with infection that died or were treated at the intensive care unit within 72 hours were also regarded as suffering from sepsis."---do you validate this statement? patients can die from other reasons but with mild infection signs. For example patients can have severe brain injury with aspiration pneumonia, but after 48 hours after ICU entry he died due to the trauma.

3. The statistical modeling lacks reference, how did you choose the cutoff points based on LOWESS for continous variables (Zhang Z, Zhang H, Khanal MK. Development of scoring system for risk stratification in clinical medicine: a step-by-step tutorial. Ann Transl Med. 2017;5(21):436. doi:10.21037/atm.2017.08.22)?

4. With multiple imputation you obtain multiple dataset, how did you combine the results? for example different datasets can give you different model coefficients and AUCs.

Reviewer #2: Strengths: This study builds on the authors' previous work in an area of critical need (i.e. improved ED-based diagnostic or screening tools for sepsis). The authors utilize statistically-sound methods to accomplish their stated goals, especially with regards to the derivation of the new decision tools (SEWS and SHEWS).

Limitations:

- I am partially uncertain why the authors decided to include a validation of RETTS in this report, in a secondary aim that is only somewhat related to the primary aim of deriving the new scores (SEWS and SHEWS) and comparing them to NEWS. This reviewer does not practice in Sweden, however, where RETTS is (as I understand it) a commonly-used system and perhaps its importance to the paper is simply the relative ubiquity of RETTS in the clinical setting of study (i.e. Swedish EDs).

-In line with above, there is some limitation in external validity outside of Sweden. This is not in any way a disqualifying issue, but probably should be mentioned in the limitations section (if page-limits allow).

- The largest concern I have is the way the sepsis 2 and sepsis 3 definitions were used here. The authors create a criterion-standard definition of sepsis which is largely based on the organ dysfunction parameters of the Sepsis 2 definition, as well as additional criteria including dying in the ICU with an infection. They then performed a sensitivity analysis comparing this definition (referred to as the combined outcome in the manuscript) with a SOFA score > 2 to adjudicate whether their definition of sepsis was concurrent with the Sepsis 3 definition.

They cite that the Sepsis 3 definition was not around at the time of sampling these patients as the reason for using their definition. I find this to be a generally unsatisfactory justification. Namely, just because sepsis 3 had not been published at time of sampling, it does not follow that it cannot be applied in a retrospective study. If the reason for using a sepsis-2 based combined definition was simply because of the authors' concerns about the validity of Sepsis 3 (a reasonable viewpoint), then I would advise them to so state. Alternatively, if feasibility of SOFA in the ED was the concern that would also be valid, but that is not stated either. Given that a sensitivity analysis using SOFA was performed, feasibility would seem to not be a major issue. Finally, throughout the article the endpoint of SOFA > 2 is used to describe the Sepsis 3 definition, however this is not accurate. The actual sepsis 3 definition is a rise in SOFA >= 2 from baseline. This helps to prevent patients with chronic disease (e.g. chronic kidney disease, cirrhosis) from being automatically classified as "septic" as soon as they hit the door (i.e. without any actual acute worsening of their end-organ function). As an example, if the definition of SOFA>=2 was used (instead of increase in SOFA >=2) it would mean every patient with stage IV-V CKD with an infection would automatically be labeled as septic in the sample even if they had no acute organ dysfunction. It is unclear if this was considered by the authors, but is eminently important since the rates of chronic renal disease were significantly higher in the septic (by combined outcome) patients in cohort A, and rates or liver disease were higher in cohort B. It may be difficult to accurately assess baseline values for SOFA with a retrospective design (i.e. to adjudicate rise vs. baseline points on SOFA) but even if so this needs to be addressed as a limitation.

6. PLOS authors have the option to publish the peer review history of their article (what does this mean?). If published, this will include your full peer review and any attached files.

Reviewer #1: No

Reviewer #2: Yes: Nicholas E Harrison

---

## [Author Response · Author response to Decision Letter 0]

4 Jan 2020

Dear Dr Ehrman,

Thank you for your letter and the comments from you and the reviewers on the manuscript entitled “Scores for sepsis detection and risk stratification – construction of a novel score using a statistical approach and validation of RETTS”.

We appreciate your engagement and interest in our work, we are pleased that the reviewers found the manuscript interesting and we are grateful for the several constructive and helpful comments. The manuscript has now been revised according to the suggestions by you and the reviewers. Our response to the specific points is given below:

Editors comment:

The first (and more significant), is the part of the definition for sepsis, as stated in lines 157-159. I wonder if this definition is overly broad, in the sense that there are many reasons for death in 72 hours or treatment in the ICU, in the setting of "infection" that are not directly sepsis. I think that further justification for this definition is required (can the charts for patients who qualified as sepsis by these criteria be reviewed?); if this is not possible, the potential ramifications of this limitation need to be discussed.

Authors reply:

Of the 233 patients with the combined outcome in cohort A, 228 had infection-related organ dysfunction, 4 were treated in the ICU and one died without organ dysfunction being registered. Of the 5 patients included in the composite outcome for sepsis but without registered organ dysfunction, 3 had culture-verified infection and 2 had probable infection. The two independent infectious disease physicians who reviewed the data attributed the death and the ICU-care for these five patients to infectious diseases. All patients with the combined outcome in cohort B had infection-related organ dysfunction registered. Perhaps this is an overly broad definition and these patients could have been regarded as not possible to classify, it would not change the result significantly, RETTS AUC 0.75 (95% 0.71-0.79), NEWS2 AUC 0.80 (95% CI 0.77-0.84). Detailed data has been added p 12, line 232-236, p 17, line 326-328 & 334-335.

Editors comment:

My second concern, although related to a secondary outcome, is about the decision to use multiple imputation for missing data in the prediction scores--I wonder if the data is actually missing not at random (MNAR). Patients who are less ill may have fewer tests/labs ordered or had VS documented less frequently (or they may be repeated less often) and thus existing values may be more deranged as they represent the sicker spectrum of the population. As such, what data does exist may not be truly representative of the entire population thereby potentially biasing the results of the multiple imputation data sets; this would be particularly true for lactate in this case. Perhaps this could explain why the two analyses (with/without) missing data produced similar results?

Authors reply:

In the imputation models we included comorbidities, outcome, systolic blood pressure, diastolic blood pressure, heart frequency, respiratory frequency, temperature, mental status, SaO2, oxygen treatment, lactate, age and HBP. We assume that the missingness for variables in the analysis model can be assumed to be missing at random (MAR) conditional on observed data in the imputation model. The MAR assumption is not testable, but becomes more reasonable with imputation models that include a wide range of characteristics, including predictors, the outcome and auxiliary variables like in our model. The result section has been changed to make it clearer, p 16, line 313-314, and p 18, line 365-366.

Reviewer 1

Reviewers comment 1. The score developed in the study requires lactate and HBP, which is difficult to obtain at the very beginning. in particular the HBP is not routinely measured. thus, the authors must discuss that the applicability of the model is limited. lactate is not routinely measured for infection but suspected sepsis.

Authors reply:

We wanted to explore different approaches for sepsis risk stratification tools i.e. a statistical approach and the addition of biomarkers and symptoms to risk stratification tools. The biomarkers, lactate and HBP, were selected for their prognostic abilities in sepsis and the fact that, even if not generally available, there exists point-of-care testing. However, even if HBP increased the performances for the new risk stratification score, it was not statistically superior to NEWS2, why the result does not support the need for availability of HBP and lactate point-of-care testing for the use in these risk stratification tools for sepsis. The manuscript has been changed in order to clarify this, p 6, line 110-111.

Reviewers comment 2. "Patients with infection that died or were treated at the intensive care unit within 72 hours were also regarded as suffering from sepsis."---do you validate this statement? patients can die from other reasons but with mild infection signs. For example patients can have severe brain injury with aspiration pneumonia, but after 48 hours after ICU entry he died due to the trauma.

Authors reply: 

As mentioned above, Of the 233 patients with the combined outcome in cohort A, 228 had infection-related organ dysfunction, 4 were treated in the ICU and one died without organ dysfunction being registered. Of the 5 patients included in the composite outcome for sepsis but without registered organ dysfunction, 3 had culture-verified infection and 2 had probable infection. The two independent infectious disease physicians who reviewed the data attributed the death and the ICU-care for these five patients to infectious diseases. All patients with the combined outcome in cohort B had infection-related organ dysfunction registered. Perhaps this is an overly broad definition and these patients could have been regarded as not possible to classify, it would not change the result significantly, RETTS AUC 0.75 (95% 0.71-0.79), NEWS2 AUC 0.80 (95% CI 0.77-0.84). Detailed data has been added p 12, line 232-236, p 17, line 326-328 & 334-335.

Reviewers comment 3. The statistical modeling lacks reference, how did you choose the cutoff points based on LOWESS for continous variables?

Authors reply: We agree and references has been added, p 9, line 193. The LOWESS curves were used to assess the relevant intervals for which dichotomization of the included continuous variables could be relevant. Each continuous variable was then dichotomized into several dummy binary variables with cut-offs spanning this relevant interval. These dummy variables were then entered into to LASSO regression. The LASSO-regression did then select the most relevant cut-offs based on statistical testing. Hence, the cut-offs for continuous variables were not manually chosen but were chosen through statistical testing in the LASSO model. 

Reviewers comment 4. With multiple imputation you obtain multiple dataset, how did you combine the results? for example different datasets can give you different model coefficients and AUCs.

Authors reply: AUC’s were calculated as medians of the pooled data and 95% CI’s for all imputed datasets. This information has now been added p 11, line 220-221, p 16, line 315-316 and p 18, line 367.

Reviewer 2

Reviewers comment 1. I am partially uncertain why the authors decided to include a validation of RETTS in this report, in a secondary aim that is only somewhat related to the primary aim of deriving the new scores (SEWS and SHEWS) and comparing them to NEWS. This reviewer does not practice in Sweden, however, where RETTS is (as I understand it) a commonly-used system and perhaps its importance to the paper is simply the relative ubiquity of RETTS in the clinical setting of study (i.e. Swedish EDs).

-In line with above, there is some limitation in external validity outside of Sweden. This is not in any way a disqualifying issue, but probably should be mentioned in the limitations section (if page-limits allow).

Authors reply: As the reviewer state RETTS is commonly used in Swedish EDs for triage and has been introduced as base for sepsis alert systems where patients, with the highest priority according to RETTS, are prioritized and treated according to sepsis bundles (Rosen qvist M. et al. Sepsis Alert – a triage model that reduces time to antibiotics and length of hospital stay Infectious diseases. 2017;49(7):507-13). Apart from the clinical setting, with limited validity outside of Sweden, we wanted to explore different approaches for sepsis risk stratification tools i.e. the addition of biomarkers and symptoms. The limitation has been addressed in the discussion, p 22, line 433-435.

Reviewers comment 2. The largest concern I have is the way the sepsis 2 and sepsis 3 definitions were used here. The authors create a criterion-standard definition of sepsis which is largely based on the organ dysfunction parameters of the Sepsis 2 definition, as well as additional criteria including dying in the ICU with an infection. They then performed a sensitivity analysis comparing this definition (referred to as the combined outcome in the manuscript) with a SOFA score > 2 to adjudicate whether their definition of sepsis was concurrent with the Sepsis 3 definition.

They cite that the Sepsis 3 definition was not around at the time of sampling these patients as the reason for using their definition. I find this to be a generally unsatisfactory justification. Namely, just because sepsis 3 had not been published at time of sampling, it does not follow that it cannot be applied in a retrospective study. If the reason for using a sepsis-2 based combined definition was simply because of the authors' concerns about the validity of Sepsis 3 (a reasonable viewpoint), then I would advise them to so state. Alternatively, if feasibility of SOFA in the ED was the concern that would also be valid, but that is not stated either. Given that a sensitivity analysis using SOFA was performed, feasibility would seem to not be a major issue. 

Authors reply: Data was prospectively gathered for vital signs and laboratory tests at arrival at the EDs and worst vital signs and laboratory results within 72 hours for cohort A. This is why we were able to perform a sensitivity analysis comparing the combined outcome with a presumed rise in SOFA score of 2 or more at arrival. We could however not calculate SOFA after arrival and found it important to evaluate the scores for prediction of sepsis and not only screening for sepsis at arrival, why we had to apply the sepsis-2 definition.

Reviewers comment 3. Finally, throughout the article the endpoint of SOFA > 2 is used to describe the Sepsis 3 definition, however this is not accurate. The actual sepsis 3 definition is a rise in SOFA >= 2 from baseline. This helps to prevent patients with chronic disease (e.g. chronic kidney disease, cirrhosis) from being automatically classified as "septic" as soon as they hit the door (i.e. without any actual acute worsening of their end-organ function). As an example, if the definition of SOFA>=2 was used (instead of increase in SOFA >=2) it would mean every patient with stage IV-V CKD with an infection would automatically be labeled as septic in the sample even if they had no acute organ dysfunction. It is unclear if this was considered by the authors, but is eminently important since the rates of chronic renal disease were significantly higher in the septic (by combined outcome) patients in cohort A, and rates or liver disease were higher in cohort B. It may be difficult to accurately assess baseline values for SOFA with a retrospective design (i.e. to adjudicate rise vs. baseline points on SOFA) but even if so this needs to be addressed as a limitation.

Authors reply: We agree, this was taken into account, but was poorly communicated in the manuscript. The text has been changed to a presumed rise in SOFA score of two or more, p 9, line 170, p16, line 321, p 18, line 372.

Additional requirements: 

The naming has been changed, p 5, line 103, p 8, line 163, p 9, line 177, line 180, p 15, line 285, p 16, line 310, 318, p 17, line 330, p 18, line 365, 375, p 18, line 381 and supporting information files.

'the regional ethical board in Lund (approval number 2010/205 and 2014/4) the regional ethical board in Bern (approval number KEK 315/14) the regional ethical board in Vancouver (approval number H11-00505).'

a. Please amend your current ethics statement to confirm that your named institutional review board or ethics committee specifically approved this study.

3. In the ethics statement in the manuscript and in the online submission form, please provide additional information about the patient records used in your retrospective study.

Specifically, please ensure that you have discussed whether all data were fully anonymized before you accessed them and/or whether the IRB or ethics committee waived the requirement for informed consent.

If patients provided informed written consent to have data from their medical records used in research, please include this information.

Authors reply to additional requirements 2 & 3: All included patients gave written informed consent in cohort A. The study is in accordance with the approval and the informed consents. The study is also in accordance with the approvals and informed consents for cohort B. In cohort B, included patients all gave written informed consent or, if unable to give informed consent, next-of-kin was asked for permission. For patients that died without being able to leave informed consent, the use of data and samples was requested at the local ethics committee. The Ethics Statement and the manuscript has been changed, p 7, line 133-135 p 8, line 152-155.

4. We note that you have a patent relating to material pertinent to this article. Please provide an amended statement of Competing Interests to declare this patent (with details including name and number), along with any other relevant declarations relating to employment, consultancy, patents, products in development or modified products etc. 

Authors reply: Bertil Christensson, Per.Åkesson, and Adam Linder are listed as inventors on a patent on the use of HBP as a diagnostic tool in sepsis filed by Hansa Medical AB WO2008151808A1. This does not alter our adherence to PLOS ONE policies on sharing data and materials. All other authors have declared no relevant conflicts of interest

5. If there are ethical or legal restrictions on sharing a de-identified data set, please explain them in detail (e.g., data contain potentially identifying or sensitive patient information) and who has imposed them (e.g., an ethics committee). Please also provide contact information for a data access committee, ethics committee, or other institutional body to which data requests may be sent.

Authors reply: The data in the study is based on patient material and since we still have a code key it is under the GDPR only considered to be pseudo-anonymized and not de-identified. Furthermore, since there are many individual variables and sensitive patient information there is a possibility that patients might be identified due to their comorbidities, patient characteristics and time of encounter. Hence, we were not granted an ethical permit to publish individual data but merely publishing data on group level. This is also clearly stated in the information to patients which the participants have signed. We consider uploading the full data set as publishing the data and due to the ethical restrictions imposed on us this is unfortunately not possible to do. In order to assure compliance with the information given to the patients, data can be shared upon request from ethics committee registrator@etikprovning.se, 0046104750800 (Swedish Ethical Review Authority) and from Cantonal Ethics Committee Zurich info.kek@kek.zh.ch. 

6. Please upload a new copy of your Supporting Information Figure as the detail is not clear. Please follow the link for more information: http://blogs.PLOS.org/everyone/2011/05/10/how-to-check-your-manuscript-image-quality-in-editorial-manager/

Authors reply: A new copy of the Supporting Information Figure has been uploaded

Authors reply: Captions for Supporting information files has been included p 23, line 464-475.

We feel that this revised manuscript is much improved compared to the previous version and hope that you will find it suitable for publication.

Again, thank you for your interest and engagement in our work.

Hoping for a positive response,

 Sincerely,

Lisa Mellhammar

---

## [Decision Letter · Decision Letter 1]

13 Jan 2020

PONE-D-19-32440R1

Scores for sepsis detection and risk stratification – construction of a novel score using a statistical approach and validation of RETTS

PLOS ONE

Dear Dr Mellhammar,

Thank you for submitting your manuscript to PLOS ONE. After careful consideration, we feel that it has merit but does not fully meet PLOS ONE’s publication criteria as it currently stands. Therefore, we invite you to submit a revised version of the manuscript that addresses the points raised during the review process.

The manuscript is markedly improved overall, but there remain some concerns about the multiple imputation models. While there may not be a strictly "correct" answer, addition of further discussion about why data was assumed to be MAR rather than MNAR and potential limitations and/or ramifications of this decision would strengthen the paper. Can you provide a reference for the sentence in bold below? If so, this would be very nice addition to the paper.

The MAR assumption is not testable, **but becomes more reasonable with imputation models that include a wide range of characteristics, including predictors,** the outcome and auxiliary variables like in our model. 

This additional text could be included in discussion of the limitations.

We would appreciate receiving your revised manuscript by Feb 27 2020 11:59PM. To enhance the reproducibility of your results, we recommend that if applicable you deposit your laboratory protocols in protocols.io, where a protocol can be assigned its own identifier (DOI) such that it can be cited independently in the future. For instructions see: http://journals.plos.org/plosone/s/submission-guidelines#loc-laboratory-protocols

We look forward to receiving your revised manuscript.

Kind regards,

Robert Ehrman, MD, MS

Academic Editor

PLOS ONE

Reviewers' comments:

Reviewer's Responses to Questions

**Comments to the Author**

1. If the authors have adequately addressed your comments raised in a previous round of review and you feel that this manuscript is now acceptable for publication, you may indicate that here to bypass the “Comments to the Author” section, enter your conflict of interest statement in the “Confidential to Editor” section, and submit your "Accept" recommendation.

Reviewer #1: All comments have been addressed

Reviewer #2: (No Response)

2. Is the manuscript technically sound, and do the data support the conclusions?

Reviewer #1: Yes

Reviewer #2: Yes

3. Has the statistical analysis been performed appropriately and rigorously? 

Reviewer #1: Yes

Reviewer #2: No

4. Have the authors made all data underlying the findings in their manuscript fully available?

Reviewer #1: Yes

Reviewer #2: No

5. Is the manuscript presented in an intelligible fashion and written in standard English?

Reviewer #1: Yes

Reviewer #2: Yes

6. Review Comments to the Author

Reviewer #1: my previous comments were adequately addressed, WELL DONE job.

The rebuttal letter is good and my comments were well addressed.

Reviewer #2: I would like to see a little more addressing the imputation, specifically the R code and more detailed demographic and outcomes data for those patients with missing data, preferably stratified by the type of data element missing. Also, a "worst-case" sensitivity analysis specifically involving lactate (since this was missing so frequently, and directly relates to a major aim), may be worth performing.

I am concerned that little in the edits seem to have substantively addressed the comments regarding imputation. The change added a sentence saying that it was assumed that data was missing at random. I found this unsatisfactory, since a large portion of reviewer and editor feedback raised was very specifically directed at needing to account for the possibility that data was missing not at random.

7. PLOS authors have the option to publish the peer review history of their article (what does this mean?). If published, this will include your full peer review and any attached files.

Reviewer #1: Yes: Zhongheng Zhang

Reviewer #2: No

---

## [Author Response · Author response to Decision Letter 1]

29 Jan 2020

Dear Dr Ehrman,

Thank you for your letter, the comments from you and the reviewers and the opportunity to revise the manuscript entitled “Scores for sepsis detection and risk stratification – construction of a novel score using a statistical approach and validation of RETTS”.

The manuscript has now been revised according to the suggestions by you and reviewer 2. Our response to the specific points is given below:

Editors comment:

While there may not be a strictly "correct" answer, addition of further discussion about why data was assumed to be MAR rather than MNAR and potential limitations and/or ramifications of this decision would strengthen the paper. Can you provide a reference for the sentence in bold below? If so, this would be very nice addition to the paper.

“The MAR assumption is not testable, but becomes more reasonable with imputation models that include a wide range of characteristics, including predictors, the outcome and auxiliary variables like in our model”.

This additional text could be included in discussion of the limitations.

Authors reply:

Discussion about limitations of multiple imputation has been added to the discussion, including the text and a reference for the statement that the MAR assumption is not testable, but becomes more reasonable within a model like ours that include several characteristics, including predictors and outcome, p 21, line 428-436 [1].

It is also elaborated in the reply to reviewer 2 below.

Reviewer 2

Reviewers comment 1: I would like to see a little more addressing the imputation, specifically the R code and more detailed demographic and outcomes data for those patients with missing data, preferably stratified by the type of data element missing. Also, a "worst-case" sensitivity analysis specifically involving lactate (since this was missing so frequently, and directly relates to a major aim), may be worth performing.

I am concerned that little in the edits seem to have substantively addressed the comments regarding imputation. The change added a sentence saying that it was assumed that data was missing at random. I found this unsatisfactory, since a large portion of reviewer and editor feedback raised was very specifically directed at needing to account for the possibility that data was missing not at random.

Authors reply:

We agree and have made an effort to address the comments on multiple imputation.

First, we have tried to clarify that even though the aims of this study were to develop a sepsis risk stratification tool based on the most predictive, minimal set of vital signs, lactate and HBP plasma levels, this was performed in cohort A, where few values for lactate were missing. 

In cohort B the new scores and RETTS were validated and lactate is not included in these scores, supporting information table IV.

To calculate worst case scenario for missing data is an approach for missing data which is a strong assumption that can give the sensitivity analyses a wide range, even with moderate number of missing outcomes. 

As mentioned previously, we assumed data to be missing at random, which is not testable, but becomes more reasonable within a model like ours that include several characteristics, including predictors and the outcome. 

The cause for that a vital sign or a laboratory value is missing is often that they were considered unnecessary according to the attending physician. The decision was then evidently made based on other available vital parameters and laboratory values and hence the probability of missing depends on other parameters present in the data set; i.e missing at random (Pr (R=0|Yobs, Ymis, �) = Pr(R=0|Yobs,�) [2].

Multiple imputation is therefore commonly used when evaluating clinical risk scores [3-5]. 

We have elaborated the discussion about limitations of multiple imputation, 21, line 428-436 and added a table on demographics and outcome of patients with missing data to the supporting information (supporting information table V).

The multiple imputation was performed in SPSS with the code below.

*set the random seed

SET MTINDEX=2000000.

*Analyze Patterns of Missing Values.

MULTIPLE IMPUTATION HR SBP DBP mental_status temperature SaO2 

 RR oxygen_treatment hbp lactate age

 /IMPUTE METHOD=NONE

 /MISSINGSUMMARIES OVERALL VARIABLES (MAXVARS=25 MINPCTMISSING=0) PATTERNS.

*Impute Missing Data Values.

DATASET DECLARE newimputeddata.

DATASET DECLARE iterationhistory.

MULTIPLE IMPUTATION HR SBP DBP oxygen_treatment RR 

 temperature mental_status SaO2 outcome hbp lactate 

 age sex comorbidities_1 comorbidities_2 

 comorbidities_3 comorbidities_4 comorbidities_5 comorbidities_6 

 comorbidities_7 

 /IMPUTE METHOD=FCS MAXITER= 100 NIMPUTATIONS=20 SCALEMODEL=PMM INTERACTIONS=NONE SINGULAR=1E-012 

 MAXPCTMISSING=NONE 

 /CONSTRAINTS age( ROLE=IND)

 /CONSTRAINTS hbp ( ROLE=IND)

 /CONSTRAINTS sex ( ROLE=IND)

 /CONSTRAINTS outcome ( ROLE=IND)

 /CONSTRAINTS comorbidities_1( ROLE=IND)

 /CONSTRAINTS comorbidities _2( ROLE=IND)

 /CONSTRAINTS comorbidities _3( ROLE=IND)

 /CONSTRAINTS comorbidities _4( ROLE=IND)

 /CONSTRAINTS comorbidities _5( ROLE=IND)

 /CONSTRAINTS comorbidities _6( ROLE=IND)

 /CONSTRAINTS comorbidities _7( ROLE=IND)

 /MISSINGSUMMARIES NONE 

 /IMPUTATIONSUMMARIES MODELS 

 /OUTFILE IMPUTATIONS=newimputeddata FCSITERATIONS=iterationhistory.

We feel that the addition of discussion on multiple imputation assumption has improved the manuscript compared to the previous version and hope that you will find it suitable for publication.

Again, thank you for your interest and engagement in our work.

Hoping for a positive response,

Sincerely,

Lisa Mellhammar

References

1. Sterne JA, White IR, Carlin JB, et al. Multiple imputation for missing data in epidemiological and clinical research: potential and pitfalls. BMJ (Clinical research ed.) 2009;338:b2393 doi: 10.1136/bmj.b2393[published Online First: Epub Date]|.

2. Buren SV. Flexible Imputation of Missing Data. CRC Press 2018 

3. Seymour CW, Kahn JM, Cooke CR, Watkins TR, Heckbert SR, Rea TD. Prediction of critical illness during out-of-hospital emergency care. Jama 2010;304(7):747-54 doi: 10.1001/jama.2010.1140[published Online First: Epub Date]|.

4. Seymour CW, Liu VX, Iwashyna TJ, et al. Assessment of Clinical Criteria for Sepsis: For the Third International Consensus Definitions for Sepsis and Septic Shock (Sepsis-3). Jama 2016;315(8):762-74 doi: 10.1001/jama.2016.0288[published Online First: Epub Date]|.

5. Sunden-Cullberg J, Rylance R, Svefors J, Norrby-Teglund A, Bjork J, Inghammar M. Fever in the Emergency Department Predicts Survival of Patients With Severe Sepsis and Septic Shock Admitted to the ICU. Crit Care Med 2017;45(4):591-99 doi: 10.1097/ccm.0000000000002249[published Online First: Epub Date]|.

---

## [Editor Report · Decision Letter 2]

3 Feb 2020

Scores for sepsis detection and risk stratification – construction of a novel score using a statistical approach and validation of RETTS

PONE-D-19-32440R2

Dear Dr. Mellhammar,

We are pleased to inform you that your manuscript has been judged scientifically suitable for publication and will be formally accepted for publication once it complies with all outstanding technical requirements.

With kind regards,

Robert Ehrman, MD, MS

Academic Editor

PLOS ONE
---

## [Editor Report · Acceptance letter]

5 Feb 2020

PONE-D-19-32440R2 

Scores for sepsis detection and risk stratification – construction of a novel score using a statistical approach and validation of RETTS 

Dear Dr. Mellhammar:

I am pleased to inform you that your manuscript has been deemed suitable for publication in PLOS ONE. Congratulations! Your manuscript is now with our production department. 

With kind regards,

on behalf of

Dr. Robert Ehrman 

Academic Editor

PLOS ONE